# Gut Bless Your Pain—Roles of the Gut Microbiota, Sleep, and Melatonin in Chronic Orofacial Pain and Depression

**DOI:** 10.3390/biomedicines10071528

**Published:** 2022-06-28

**Authors:** Łukasz Lassmann, Matteo Pollis, Agata Żółtowska, Daniele Manfredini

**Affiliations:** 1Dental Sense Medicover, 80-283 Gdańsk, Poland; 2Department of Medical Biotechnology, School of Dentistry, University of Siena, 53100 Siena, Italy; matteo.pollis@gmail.com (M.P.); daniele.manfredini75@gmail.com (D.M.); 3Department of Conservative Dentistry, Faculty of Medicine, Medical University of Gdańsk, 80-210 Gdańsk, Poland; agata.zoltowska@gumed.edu.pl

**Keywords:** gut microbiota, temporomandibular disorders, orofacial pain, bruxism, melatonin, sleep, depression, gastroesophageal reflux disease, irritable bowel syndrome, inflammation

## Abstract

**Background.** Increased attention has been paid to the gut–brain axis recently, but little is known so far regarding how this translates into pain susceptibility. **Aim.** The aim of this review is to determine whether gastroenterological disorders and sleep disorders (directly or indirectly) contribute to an increased susceptibility to depression and chronic orofacial pain. **Method.** A search was performed in the U.S. National Library of Medicine (PubMed) database in order to find studies published before 19 December 2021. We used the following terms: gut microbiome, OR sleep quality, OR melatonin, OR GERD, OR IBS, AND: depression OR chronic pain, in different configurations. Only papers in English were selected. Given the large number of papers retrieved in the search, their findings were described and organized narratively. **Results.** A link exists between sleep disorders and gastroenterological disorders, which, by adversely affecting the psyche and increasing inflammation, disturb the metabolism of tryptophan and cause excessive microglial activation, leading to increased susceptibility to pain sensation and depression. **Conclusions.** Pain therapists should pay close attention to sleep and gastrointestinal disorders in patients with chronic pain and depression.

## 1. Introduction

Among the orofacial pain conditions, the most prominent are the temporomandibular disorders (TMDs) [1]. These disorders affect about 10–15% of the population at a clinically relevant level [2], with 90% of cases reporting pain in the masticatory muscles and tenderness to palpation [3,4].

TMDs are often managed in the dental office; however, they are complex medical conditions, being unrelated to the features of dental occlusion and requiring an interdisciplinary approach. A current gap in TMD practice is the difficulty in training clinicians with respect to the need for evaluation of possible vulnerabilities that are not directly associated with the masticatory organ [5].

Recent suggestions have highlighted the importance of multidisciplinary rehabilitation in the management of chronic pain [6] and gastrointestinal diseases, including gastroesophageal reflux disease (GERD) and irritable bowel syndrome (IBS). It turns out that both GERD and IBS are separately associated with an almost three times higher risk of TMD [7], and the interplay between these diseases is at least partially connected with GERD anti-acidic treatment, which significantly disrupts the gut microbiota—the population of micro-organisms that colonizes the intestines. These micro-organisms produce metabolites, such as serotonin, dopamine, gamma-aminobutyric acid (GABA), and short-chain fatty acids (SCFA), which affect the activity of the central nervous system (CNS) [8,9,10]. This action may modulate many types of chronic pain, including visceral, inflammatory, headache, and neuropathic pain [11]. In addition, some experimental studies have shown that sleep deprivation increases pain sensitivity and leads to hyperalgesia [12,13]. Studies have also confirmed the complete normalization of pain sensation after so-called “recovery sleep” [14], which seems to be important in terms of the possible TMD–sleep connection [15,16,17,18,19]. Moreover, sleep disorders increase susceptibility to stress [20] and have a negative effect on the intestines [21], which, in turn, enhances the stress response [22]. For these reasons, it seems probable that an interaction between sleep disturbances, gut microbiota alteration, and the chronicization of pain may exist. A considerable amount of evidence has contributed to extending the knowledge around this interesting phenomena in recent years; however, the complexity of the interactions and the multiple pathophysiological processes involved make it difficult to understand. Therefore, the aim of this review is to explore the complex picture of individual vulnerability, genetics, and developmental disorders, which is likely to also include factors related to sleep and the gastrointestinal system. 

To the best of our knowledge, this is the first paper presenting possible TMD–pain models involving the inclusion and discussion of quality of sleep, psychological factors (i.e., somatization, depression, anxiety, and PTSD), and gastrointestinal factors (i.e., GERD, IBS, and gut microbiome). Furthermore, several substances that appear to have a positive effect on each of these risk factors are presented. We devote an entire section to the most important of these substances—melatonin. Finally, a possible sequence of events leading to chronic orofacial pain will be presented, with a possible answer to the question of why women suffer more.

## 2. Search Strategy and Selection Criteria 

A search was performed in the U.S. National Library of Medicine (PubMed) database, in order to find studies published before 19 December 2021. We used the following terms to identify possible chronic orofacial pain contributors and their relationships: gut microbiome, OR sleep quality, OR melatonin, OR GERD, OR IBS. These were combined with terms to determine the outcomes of interest: depression OR chronic pain. Inclusion criteria were limited to: (1) relevant papers describing the relationship between at least two of the terms mentioned; (2) papers with statistically significant outcomes; (3) papers focused on the etiology of chronic pain and depression; and (4) studies written in English. The primary author (L.L) independently reviewed the titles and abstracts of all articles, followed by a full-text eligibility check. Any studies not fulfilling the inclusion criteria were excluded from further evaluation, resulting in the eventual qualification and inclusion of 142 out of 1180 papers. Given the large number of papers retrieved in the search, their findings are described and organized narratively in the following. 

## 3. Sleep–Psycho–Pain Axis

Modern sleep research has shown that sleep disruption is linked to the development and progression of anxiety disorders [23,24,25]—the most common mental illness worldwide at present [26].

Interestingly, it has been reported that REM sleep acts as a potential night-time psychotherapist for anxiety and stress [20]. Neuroimaging studies have revealed significant activity increases during REM sleep in emotion-related regions [27,28,29], where these changes are regulated by striking neurochemical alterations [30,31,32]. Perhaps most remarkable is a substantial reduction in the level of noradrenaline during REM sleep [31,33,34,35,36], the lowest at any time during the 24 h period. In this calm, noradrenalin-free environment, REM sleep involves very emotional and often very aggressive dreams [37,38,39], serving to transform emotional memories into memories that are no longer emotional. Therefore, after a good night of sleep, something that previously stressed us may no longer arouse our fears [36]. It is this mechanism that fails in the case of PTSD, where the brain cannot clear itself from excessive amounts of noradrenaline [37].

Non-rapid eye movement slow-wave oscillations (NREM-SW) also offer an ameliorating, anxiolytic benefit [38], but the role of NREM sleep can be considered separate from that of REM sleep in the regulation of emotions [39]. Specifically, (NREM-associated) anxiety is a state that operates across a time-frame of hours, while (REM-associated) emotional reactivity is considered to be a short-term, acute process that begins and ends within a time frame of milliseconds to minutes [40].

Studies in animals [41] and humans have established that total [42], partial [43], and selective sleep deprivation [44,45] can lead to increased pain. It is particularly the lack of the deep phase of non-REM sleep that increases pain sensitivity [46]. One of the associated mechanisms is the lowering of pain thresholds [47].

Sleep disorders and pain susceptibility can aggravate each other, suggesting a bidirectional and reciprocal relationship [48]; however, recent studies have pointed toward a stronger and more consistent unidirectional effect of sleep causing pain exacerbation [49,50,51].

Interestingly, sleep can be also disrupted by inflammatory processes. During the up-regulation of pro-inflammatory cytokines (including IL-1 and TNF), NREM sleep is increased [52,53], but also fragmented [54], while REM sleep is suppressed [55]. As a consequence, the lack of quality REM and non-REM sleep worsens body regeneration and increases pain sensitivity, which is believed to occur through dysregulation of the hypothalamus–pituitary–adrenal (HPA) axis and nociceptive perception [56]. Sleep is also altered during clinical conditions, such as depression [57]. Interestingly, pro-inflammatory cytokines and their mediators have also been shown to be associated with depression [58,59,60].

## 4. Inflammation–Psycho–Pain Axis

It is worth mentioning that not all cytokine depressive responses are equal, as different inflammatory profiles are associated with different sub-types of depression [61,62,63]. However, in general, it is known that pro-inflammatory cytokines are increased in major depressive disorder (MDD) [64,65,66,67], with the most remarkable increases in IL-6, TNF, and C-reactive protein (CRP) [68,69,70]. It has also been recently reported that IL-17 produced by gamma delta T-cells (gdTcells) may be a key link between the immune system and its effects on the body and mind, with particular emphasis on anxiety [71]. 

It is currently well-documented that depression is a strong predictor of orofacial pain and increased risk for the development of TMD [72,73,74,75,76]. This comorbidity may be due to overlapping pathways. Regions in the brain that are responsible for emotions send projections to pain modulation structures in the brainstem, which could explain why depression (a negative emotion) is often accompanied with an intensified pain response [77]. 

In 1977, Engel proposed the “biopsychosocial model” to address chronic pain in medicine [78]. This novel point of view led to the creation of the DC/TMD classification, including both a physical (Axis I) and a psychosocial (Axis II) appraisal [79]. Interestingly, the prognosis for patients with TMD is influenced more by Axis II factors than Axis I factors [80,81], emphasizing the importance of the psychological aspect in the pathogenesis of TMD. Knowing this, it would be reasonable to ask what the biological reasons for depression, anxiety, and increased pro-inflammatory cytokines are, in general. It turns out that our intestines—considered by some to be our “second brain” [82]—are becoming the prime suspect.

Tryptophan derived from dietary proteins can be metabolized into different substances, such as indole derivatives, which can induce anxiety. Tryptophan can also metabolize through the kynurenic pathway which, in microglia, is used to produce neurotoxic quinolinic acid (Quin), whereas astrocytes generate kynurenic acid (Kyna), which reduces neuronal excitability [83]. The third pathway of tryptophan metabolism leads to the production of serotonin, which has many positive effects in the CNS, but may also be a pain-inducing mediator that transmits pain signals directly to the vagus nerve [84]. Melatonin can be produced both in the gut, where it improves the function of microbiota, or in the pineal gland, where it regulates the circadian rhythm. Within the gut, bacteria such as *Lactobacillus* spp., *Bifidobacterium dentium*, and *Bifidobacterium* spp. [85] can also produce the inhibitory neurotransmitter GABA, which causes desensitization [86,87]. (Adapted from “Gut–Brain Axis”, by BioRender.com Retrieved from https://app.biorender.com/biorender-templates) (accessed on 1 January 2022).

## 5. Gut–Psycho–Pain Axis

The emerging role of the gut microbiota in neurological and psychiatric disorders has recently been demonstrated [88]. The whole cascade often starts with emotional stress, which can increase intestinal permeability, allowing bacteria to move across the intestinal mucosa [89] and causing visceral hyperalgesia (Figure 1). This happens as corticosteroids increase intestinal permeability through decreased levels of claudin-1, occludin, and zona occludens-1 [21]. Knowing that TMD has been associated with gastrointestinal disorders, especially IBS [90,91], it seems reasonable to ask whether the gut microbiota—in addition to its influence on the psyche—also has a direct influence on the perception of pain.

It turns out that increased gut permeability, caused by cortisol, opens the gate for bacterial cell-wall components (e.g., lipopolysaccharides; LPS) that bind to pattern recognition receptors (PRR) expressed on immune cells and sensory neurons located in dorsal root ganglia [97,98]. This mechanism is considered to be an important contributor to peripheral sensitization [99]. Another important cause of sensitization comes from dietary tryptophan (Trp) and its metabolites. Interestingly, the microbiota play a key role in this metabolism [100] (Figure 2), which has three main pathways:Production of serotonin (5-HT). Surprisingly, 90% of serotonin is produced by Enterococcus spp. in the gut. Decreased serotonin levels are associated with depression and, as recently hypothesized, with severe sleep bruxism [101]. It is also directly connected with a lack of its derivative—melatonin—as discussed in the following section. It turns out that inflammatory processes in the guts, as well as stress, cause the depletion of serotonin by changing the metabolic pathway of Trp [102] to the second one—the kynurenine pathway (KP) [103].Kynurenine may be both good and bad for pain perception, as it may be further metabolized through two different pathways: either to kynurenic acid (KYNA), which reduces neuronal excitability [104,105], or quinolinic acid (Quin), which is neurotoxic [83].The third metabolic pathway for Trp involves the production of indoxyl sulfate, which is associated with anxiety. Strikingly, the gut microbiota is exclusively responsible for the conversion of tryptophan to indole derivatives [106]. This provides further evidence that gut microbiome disruption can directly contribute to neuropsychiatric disorders [107], and that the link between our emotional state and microbiota is bidirectional [108,109]: stress disrupts the gut microbiome which, in turn, can cause stress.

Growing evidence has also suggested that disturbance of the gut microbiota significantly influences microglia maturation [110,111,112], the main functions of which are self-renewal to maintain CNS homeostasis and rapid responses to damage or infection [113]. However, prolonged activation of microglia contributes to disruption of the synthesis, reuptake, and release of neurotransmitters [114,115,116], as well as excessive production of pro-inflammatory cytokines, and can result in increased synaptic glutaminergic neurotransmission and decreased GABAergic synaptic neurotransmission [117,118,119,120]. This, in turn, contributes to the development of central sensitization and affects mood and cognition [121], with a particular emphasis on depression [122,123,124].

GERD affects the gastro-intestinal tract, but seems to also play a large role in terms of gut dysbiosis and pain. As GERD is associated with teeth grinding or clenching [125,126,127,128,129] and an almost three times higher risk of TMD, it is important to understand whether it is the disease itself or the side-effects of proton pump inhibitor (PPI) treatment that contributes to orofacial pain. Indeed, recent evidence has shown that pharmacological treatment of acid reflux could be more harmful than the disease itself. Some drugs for GERD treatment can negatively influence many intestinal processes [130], impair magnesium absorption [131,132,133], and contribute to malabsorption of vitamin B12 [134,135,136], which is manifested by neuropathy and mood disorders including personality change, psychosis, and emotional lability [137]. In addition, as stomach acid levels decline, more bacterial types can survive in the stomach and enter the small intestine, which can cause significant changes in the gut microbiota composition [138,139,140,141,142]. Moreover, by disrupting the bacterial flora, these drugs have been associated with problems such as irritable bowel syndrome [143,144], small intestinal bacterial overgrowth (SIBO) [145,146,147], and even stomach cancer [148,149,150,151]. The study of such possible negative consequences of PPIs has led to a search for alternative methods to treat GERD, such as melatonin supplementation [152,153,154,155].

Thus, knowing the possible metabolic mechanisms leading to sensitization, we should search for therapeutics that are anti-inflammatory, improve the function of the gut microbiome and, consequently, improve sleep and mood. There are at least five products that seems to fulfill these expectations, in terms of orofacial pain:4.Probiotics (e.g., *Bacteroides fragilis*) can correct some of the changes related to increased gut permeability [156]. Probiotic supplementation has also shown promising results, with reductions in anxiety and depression [157,158,159,160] through direct and indirect mechanisms of action; for example, local stimulation of the vagus nerve [161,162,163,164,165], which reduces the activity of the sympathetic nervous system. Considering the potential beneficial effects of probiotics on mental health, they are also referred to as “psychobiotics” [83].5.Omega-3 fatty acids show anti-inflammatory effects on LPS-stimulated microglia [166], induce an increase in several short-chain fatty acid-producing bacteria species [167], and help to maintain intestinal wall integrity. Thanks to those properties, they have been shown to be effective in the treatment of gut dysbiosis [168] (disruption to the microbiota homeostasis), depression [169], neuropathic pain [170] after neurotrauma [171], joint pain associated with rheumatoid arthritis, and inflammatory bowel disease [172].6.Resveratrol is antioxidative, anti-inflammatory [173], and improves the gut microbiota [174]. Recently, resveratrol has been used to alleviate temporomandibular joint inflammatory pain by recovering disturbed gut microbiota. An interesting observation is that the systemic administration of resveratrol restored reduced Bacteroidetes and Lachnospiraceae while attenuating nociception in TMJ-inflamed mice, where the antinociceptive effect was mimicked by fecal transplantation from inflamed animals receiving resveratrol treatment [175].7.Short-chain fatty acids (SCFAs), which are derived from bacterial fermentation of dietary fiber in the gut [176], play important roles in regulating microglia morphology and function. SCFAs may act as important mediators derived from the gut microbiota for regulation of pain through receptor-mediated mechanisms, epigenetic regulation mechanisms, or both [177,178,179,180].8.The last therapeutic that should be discussed in this review is melatonin, which deserves a separate section.

## 6. Role of Melatonin in Pain, Sleep, and Inflammation

Melatonin, acting as a pleiotropic hormone, is released from the pineal gland and extra-pineal tissues, and plays a critical role in regulating the circadian rhythms [181]. Melatonin is largely produced in the intestines, where it reaches concentrations 400 times greater than that in the pineal gland and up to 100 times greater than that in the blood [182,183]. Initially, it was described as a sleep hormone, as it is secreted in the dark and induces sleep; however, it is now widely appreciated that it presents a wide array of activities (see Figure 3), encompassing anti-oxidant, anti-inflammatory, anti-apoptotic, anti-sympathetic nerve activation, endothelial cell preservation, neuroprotection, hepatoprotection, immunomodulation, thermoregulation, mood, and sexual behavior modulation [184,185,186,187,188]. 

Melatonin’s effect on pain is consistent with previous clinical and experimental data [196,197,198]. Regarding a low intensity of pain perception during the night, the possible analgesic effect of high melatonin during the night has been proposed as an associated mechanism [199]. Zhu et al., in a systematic review and meta-analysis of 19 randomized controlled trials using melatonin for various types of pain, reported a significant reduction in pain [200]. A recent meta-analysis of randomized, double-blind, and placebo-controlled trials concluded that melatonin might be used for the treatment of chronic pain, specifically endometriosis, IBS, and migraines [201].

Thus, melatonin may decrease pain by improving sleep through circadian rhythm normalization [202], but also through its own action on melatonin receptors and several neurotransmitter systems [193]. Animal models have also demonstrated that the suppression of melatonin secretion due to sleep deprivation can increase glial activation and aggravate neuropathic pain [203]. Moreover, disrupted melatonin secretion has been related to clinical symptoms in major depression and fibromyalgia patients [204].

Even national organizations on sleep research have started to recommend melatonin for insomnia symptoms, as well as in mood disorders, fibromyalgia, irritable bowel syndrome, functional dyspeptic syndrome, and temporomandibular joint dysfunction [192].

A recent double-blind, randomized, placebo-controlled study demonstrated that melatonin produces a reduction in overall pain, compared with placebo, in the treatment of myofascial TMD pain. In addition, it seems that the effect of melatonin on pain may be independent of the improvement in sleep quality. This conclusion is clinically relevant, as it suggests that its use does not need to be restricted to patients with pain and sleep disturbances [205].

There has been only one single case study relating bruxism with melatonin. A 7-year-old girl with sleep bruxism and sleep talking was managed with melatonin at 1.5 mg/day, with good results in two weeks and no adverse effects [206].

Additionally, a recent systematic review and meta-analysis of clinical trials indicated that interventions longer than 12 weeks and at a dosage of ≥10 mg/day were more efficacious in attenuating IL-6 and TNF-α levels, showing that long-term interventions with high doses of melatonin are required to effectively reduce inflammation [207]. From the available studies, doses of 10 mg (increasing blood levels up to 60-fold) [208] appear to be safe [209,210] and non-addictive [211].

The influence of melatonin on sex hormones has also become clearer than in the past. In men, clinical trials have indicated no differences in the hormonal synthesis of luteotropic hormone (LH), follicle-stimulating hormone (FSH), and testosterone after melatonin administration [212,213]. To the contrary, in women, treatment with melatonin decreased luteotropic hormone, estradiol, and progesterone [214,215,216]. Moreover, its potential to act directly on the epithelial mammary cells designates melatonin as a selective estrogen enzyme modulator [217,218]. The ability of melatonin to modulate estrogen appears to be particularly important in considering the sexual dimorphism of pain perception.

## 7. A proposed Cascade of Events Leading to Orofacial Pain and Gender Predisposition

There are multiple vulnerability factors that may perpetuate pain and facilitate the transition to chronic conditions, with a certain degree of unpredictability at the individual level [219].

An example case of a female patient with a history of PTSD can be used to depict a possible cascade of events. Importantly, the described cascade is by no means limited to people with PTSD which, in this case, is simply used for the sake of discussing the possible cascade of events and sex differences for risk factors.

PTSD is associated with a 2.56-fold higher risk of TMJ pain and 3.86-fold higher risk of muscle pain, with both types of pain being significantly higher for women [220]. Moreover, the risk of PTSD across the lifetime is also significantly higher for women than for men [221,222]. Interestingly, major depression is nearly two times more likely to occur in women than men [223], and women also show a higher inflammatory response to acute stress than men [224]. Based on the results from an 11-year follow-up study on Finnish adults, the effect of depressive symptoms on temporomandibular pain was also more direct in women [77]. Overall depression, anxiety, and somatoform disorders are all more prevalent in women than in men; however, the specific biological mechanisms contributing to such sex differences have only recently been discovered. The serotonin transporter (SERT), encoded by the SLC6A4 gene, turned out to be one of the causative mechanisms by which women exhibit an increased prevalence of somatic symptoms [225].

During PTSD, due to chronic stress, the gut epithelial layer become more permeable, leading to an increased movement of endotoxins and resulting in a low-grade inflammation and progressive tendency to lowered psychological mood. It is likely that, for this reason, trauma in childhood is associated with elevated levels of pro-inflammatory cytokines in later life [226]. Stress and inflammatory processes change the metabolic pathway of tryptophan towards the kyneurenic pathway. This, in turn, decreases levels of serotonin, with all of the possible consequences associated with depression, sleep bruxism [101], and dysregulation of the circadian rhythm due to a lack of melatonin. 

Women are also 41% more likely than men to experience insomnia [227]. Moreover, acute sleep loss leads to alterations in inflammatory gene expression [228], which, in women, showed greater up-regulation compared to that in men [229]. Inflammatory cytokines may directly disrupt sleep, but may also be associated with depression, which changes the sleep composition even further. As a consequence, patients lacking REM sleep are likely to be more emotionally reactive. Their NON-REM sleep is fragmented, causing higher anxiety, fatigue, and increased pain sensitivity.

In short, the worse the sleep, the more the pain and stress. Conversely, in the case of our hypothetical patient with PTSD, the higher the stress level, the more extreme the gut dysbiosis and the worse the sleep quality.

In these conditions, it is quite easy to get used to substances that give relief, as a chronic lack of sleep is conducive to addiction [230,231,232,233]; in particular, PTSD significantly increases the risk of alcohol use disorder [234]. Generally, individuals who are particularly sensitive to stress drink alcohol to fall asleep [235]; however, this habit makes sleeping difficult and worsens sleep apnea [236], especially if it was already present. Indeed, alcohol before bedtime causes sleep fragmentation and, notably, additionally reduces REM sleep [237]. Notwithstanding, if the patient is diagnosed with depression and treated with antidepressants, such as SSRIs, these drugs may carry the risk of depriving someone of REM sleep and its beneficial effects on the regulation of emotions [238]. Interestingly, daily sleepiness after poor quality sleep stimulates over-eating with unhealthy food [239], which may affect the tone of the esophageal sphincter leading to GERD. Reflux symptoms affect women more than men [240]. Additionally, physiological stress increases the perception of heartburn and aggravates GERD symptoms, increasing the need for treatment [241].

During GERD treatment, PPI-driven gastric hypochlorhydria can modify the composition of the gut microbiota, adding to the stress-mediated changes. As described in the preceding sections, dysbiosis can directly modulate the neuronal excitability, contributing to many types of chronic pain. This may partially explain the findings that both GERD and IBS are associated with a 3 times higher risk of TMD. However, this mechanistic explanation should be considered as part of a bigger picture, knowing the relation between GERD and psychological disorders [242,243,244] and that between psychological factors and the onset and persistence of TMD [245]. Sex hormones also seem to play an important role in this case. Estradiol directly increases the number of Proteobacteria species and decreases Prevotellaceae in females, causing increased LPS (lipopolysaccharide) and decreased SCFA production, respectively, thereby increasing the risk of mental disorders in the pubertal and reproductive phases [246,247,248]. Moreover, oral contraceptives and ovariectomy are also associated with changes in the gut microbiota [249]. All of these mentioned risk factors are higher for women (Figure 4), likely mediated by genetic predispositions and the fluctuation of sex hormones. 

Therefore, it should not be surprising that the occurrence of malocclusion, occlusal interferences, and missing teeth is nearly equal for males and females [250], while it is known that the risk of TMD in women is more than twice as high [251], as well as the need for treatment. 

As a final result, patients may take several medications for IBS, GERD, depression, anxiety, insomnia, pain, neuropathy, inflammation, and muscle soreness, contributing to the perpetuation of an interactive viscous circle of chronic diseases (Figure 5). This may be a proposed mechanism to link the various conditions/phenomena. In this vicious cycle, it is not possible to state with certainty where it begins yet; at any rate, it is likely that the primary condition differs at the individual level. Based on this, future research on the interactions of factors implicated in the development of the gut–brain axis and their influence on pain, mood, and sleep modulation is recommended.

## 8. Conclusions

From this review emerged the fact that inflammation, through disruption of the sleep cycle, may worsen body regeneration and increase pain sensitivity, anxiety, and stress. This, in turn, increases intestinal permeability and disrupts the microbiota, leading (directly and indirectly) to sensitization of the central nervous system, nutrition malabsorption, and hyperactivation of microglia, possibly contributing to many types of chronic pain, including visceral, inflammatory, headache, and neuropathic pain. The gut microbiome can be also negatively altered by GERD, especially during treatment with PPIs. Inflammation and disruption of the intestinal microbiome alter the metabolism of tryptophan and its important derivatives, serotonin and melatonin, both of which seem to be crucial regulators of pathophysiology in the treatment of chronic orofacial pain. GERD, IBS, sleep disorders, anxiety, depression, PTSD, hyperalgesia, and somatization are all more prevalent among women than men, and so are TMDs. This might further support the hypothesis of the existence of a Gut–Sleep–Psycho–TMD axis.

Key Findings

Inflammation negatively influences the sleep cycle, leading to a pre-disposition to higher pain sensitivity, anxiety, and stress and, as consequence, to intestinal permeability and disrupted microbiota;Bacterial dysbiosis leads (directly and indirectly) to sensitization of the central nervous system, possibly contributing to many types of chronic pain;Inflammation and disruption of the intestinal microbiome alter the metabolism of serotonin and melatonin;GERD, IBS, sleep disorders, anxiety, depression, PTSD, hyperalgesia, and somatization are all more prevalent among women than men, and so are TMDs.

## Figures and Tables

**Figure 1 biomedicines-10-01528-f001:**
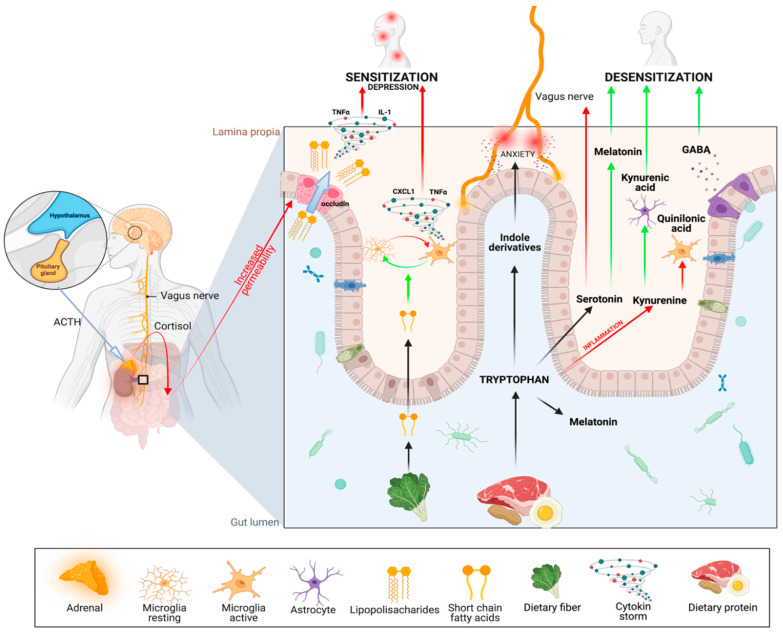
Communication between the gut microbiota and the central nervous system (CNS) in orofacial pain.Within the nervous system, stress can activate the HPA (hypothalamus–pituitary–adrenal) axis response, triggering the release of adrenocorticotrophic hormone (ACTH), which then initiates the synthesis and release of cortisol. Cortisol, in turn, affects intestinal barrier integrity by decreasing occludin levels. Several types of cells in the brain (e.g., microglia, astrocytes) are able to receive signals from the periphery, including the gastrointestinal tract [92,93,94]. Activation of these cells by lipopolysaccharides contributes to the development of depression and neuritis, which is one of the major mechanisms underlying the central sensitization associated with chronic pain [95,96]. Short-chain fatty acids derived from dietary fiber play an important regulatory role in activating microglia, thus protecting from sensitization.

**Figure 2 biomedicines-10-01528-f002:**
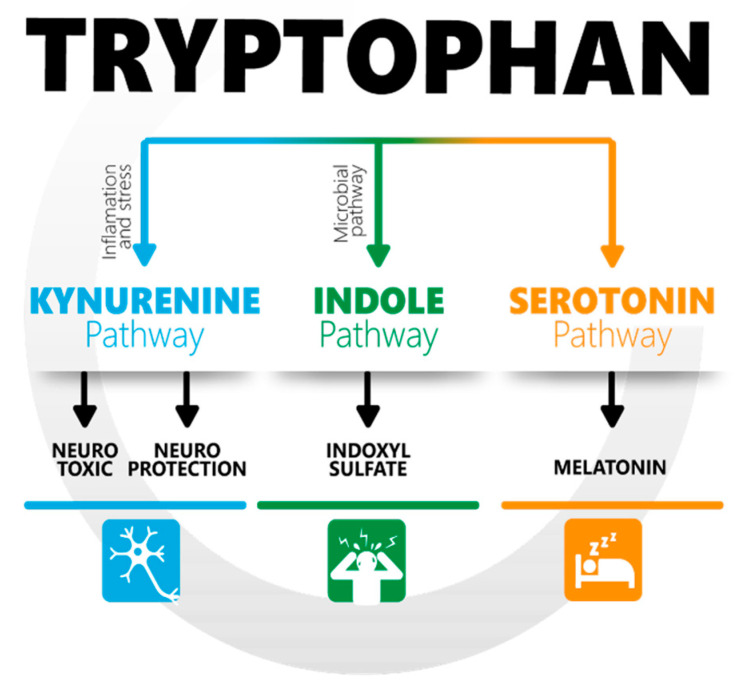
Tryptophan is processed by three different metabolic pathways, depending on the presence or absence of inflammatory process, stress, and actual requirements of the host (own resources).

**Figure 3 biomedicines-10-01528-f003:**
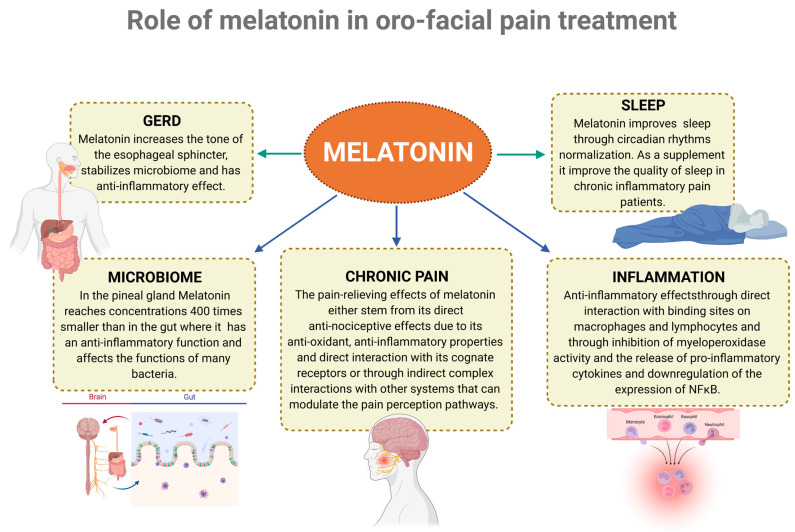
Melatonin improves the function of many processes that modulate or predict the presence of orofacial pain (Created with BioRender.com). GERD [189], Microbiome [190,191], Chronic pain [192,193], inflammation [194,195], sleep [192].

**Figure 4 biomedicines-10-01528-f004:**
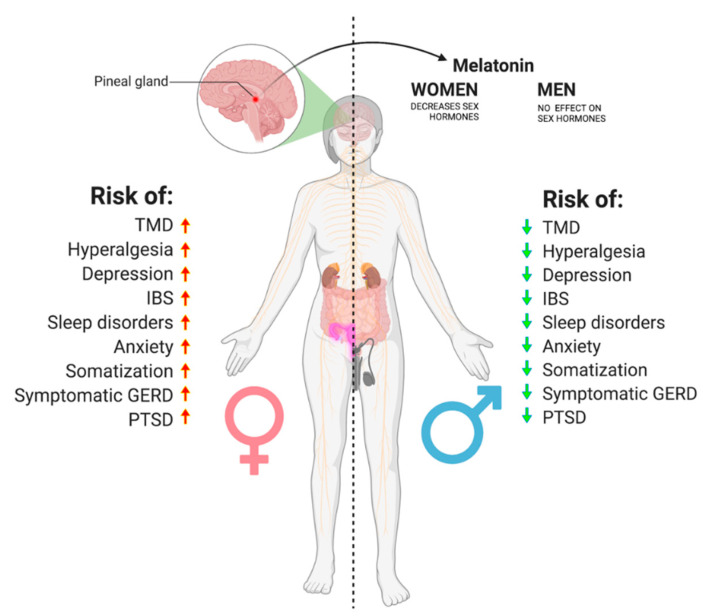
All of the TMD risk factors discussed in this review appear to be much more common in women. Moreover, melatonin seems to affect female sex hormones, while not influencing male testosterone (Created with BioRender.com) (accessed on 1 January 2022).

**Figure 5 biomedicines-10-01528-f005:**
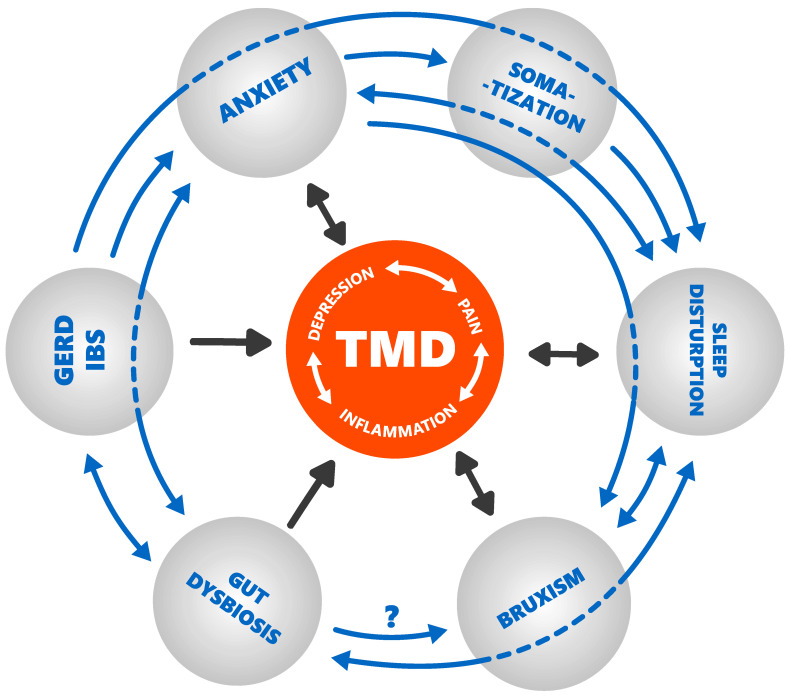
Vicious cycle of chronic diseases leading to chronic temporomandibular disorders (own resources).

## Data Availability

Not applicable.

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
