# Peer review of "Gut Bless Your Pain—Roles of the Gut Microbiota, Sleep, and Melatonin in Chronic Orofacial Pain and Depression"

_biomedicines, 2022, doi:10.3390/biomedicines10071528_

Round 1

Reviewer 1 Report

The article entitled “Gut bless your pain – the role of gut microbiota, sleep and melatonin in chronic orofacial pain” is a narrative review presenting literature evidence to support the hypothesis of the gut-sleep-psycho-temporomandibular disorders axis. 

The article concerns an interesting topic and contributes to the field quite well. Unfortunately, it is poorly written and, as a consequence, quite hard to follow by the reader. The major drawback is the use of English, which, in my opinion, must be corrected by the professional service of a native English speaker. Furthermore, the overall merit of the presented review is acceptable, however, could be improved. According to the SANRA scale (the assessment of the quality of narrative reviews), the review achieved 8 points (with 12 points maximum). The article could be improved within criteria 1, 3, 5, and 6. Justification of the article’s importance should be presented more clearly to the reader, the description of the literature search should be more detailed (inclusion criteria), and scientific reasoning should mention study design/levels of evidence of cited studies. Moreover, data in the review should be presented more adequately - without repeating data in the text, and the Figures. Figures should present more concrete data, not only a general view of described mechanisms.

Another major objection concerns the abstract of the paper which is almost exactly the same as the conclusion. It seems highly unprofessional and does not bring any new observations for the reader in the conclusions section. Abstract provides only some general background and lacks basic information, such as methods (as the narrative review also includes search methodology), results, and conclusions of the article. In my opinion, both sections – abstract and conclusions – must be rewritten.

Other comments are listed below:

1. Abbreviations are not explained in the abstract. 

2. Figure 1 seems to have an unfinished description.

3. Figure 3 does not provide any new information than the following text. I believe it should be either presented on the figure and only commented on in the text or the figure should be deleted and then the description could be retained.

4. Figure 4 gives misleading information. For men, green arrows point down, which usually indicates the lower risk of listed conditions. However, the upper part of the figures states that there is no effect of melatonin on men's sex hormones. Which is it? No effect or lower risk? The Figure should be reconsidered and corrected. 

5. Figures (all of them) do not provide information on the source of the used graphics or the tool used to prepare them (such as biorender etc.), please address it and add this information where appropriate.

6. Title of section 6 seems incomplete. It should clearly state what events the authors mean and the differences between what would be discussed. Please correct it. 

7. The overall use of English is poor and must be corrected throughout the manuscript.

·       Past simple should be used throughout the text. 

·       Punctuation must be corrected. 

·       The numerous phrases are wordy and unnecessary and the choice of words is incorrect. 
Examples: 
- “This contributes to create a potential circle of interacting phenomena that are worthy of further exploration” (line 48) – it could be changed to a more professional and clearer sentence “The interaction of mentioned phenomena should be further explored”. 
- “melatonin will be devoted to a whole section” (line 55)
- “the lowest of any time during the 24hr period” (line 74)
- “of pain” (line 85)
- the phrase “it has also been reported that…” is unnecessary (line 103) and could be shortened to “IL-17 produced by…”
- “that are” (line 110) should be deleted
- “from both” in the figure 1 description
- “it happens because” (line 125)
- “is considered to be important” (line 131)
- “has a key role” (line 133) correct phrase should sound “plays a key role”
- line 134 is very misleading “production of serotonin, which surprisingly 90% comes from the gut, decreased…” might instead sound “Serotonin production. 90% of serotonin is produced by Enterococcus spp. in the gut. Decreased…”
- “cytokines and can” (line 155)
- “there is also one particular disease” (line 162) should be removed and the sentence should start from “GERD, affects…”
- “can cause also” (line 170)
- “search for therapeutics are” (line 175)
- “recent studies show that (scfas) may be important…” (line 194) – parentheses are unnecessary and the sentence should be shortened, for example, to “SCFAs may act as important mediators…”
- “the ability melatonin to modulate” (line 258) lacks “of”
- “this complexity comes from the fact that there are” (line 264) may be deleted
- “an example case of” (line 267)
- “this may be a reason to explain” (line 281) should be removed
- lines 288-289 should be corrected, for example, to “As a consequence, patients lacking REM sleep are being more emotionally reactive. Their NON-REM sleep is fragmented causing higher anxiety, fatigue, and increased pain sensitivity.
- “also in this case” (line 310)
- “all the above mentioned” (line 319) should be deleted
- “the vicious circle” (line 329) – there is no such phrase in English; it should be “the vicious cycle”
- “can be seen as a loop without a clear head and tail” (line 330) is unnecessary as readers comprehend the meaning of the “circle/cycle” in the context
- “researches” (line 331) should state either “researchERS” as in people or “research” as there is no plural form of a mass noun

·       The frequent use of clearly Polish syntax makes the article hard to follow and understand. Examples: 
- “a potential nighttime psychotherapist” (line 71)
- “thanks to those properties they proved” (line 185
- all phrases containing “it is”, “this”, “so”, “that”, “it turns out that”, “there is/are”, and “these” are used too frequently and when placed at the beginning of the sentence set an unprofessional tone of the article and should be changed and/or deleted.  
- “our example patient” (line 290) – was it in fact YOUR patient?

Author Response

Dear reviewer,
Thank you for such in-depth analysis and constructive comments.
I will try to answer each comment.
- article selection criteria have been improved.
- "Figures should present more concrete data" - the aim of my work is to reach a wide group of specialists, including dentists who need clear diagrams devoid of excessive detail in order to better understand these connections.
- abstract has been rewritten to meet the requirements
- "Abbreviations are not explained in the abstract." - there are so many abbreviations that I decided to put them at the end of the article. Where should I mention where the reader should look for them?
- "Figure 1 seems to have an unfinished description." - the text frame was shortened by mistake, it has already been corrected
- Figure 3 provides a pictorial summary of the role of melatonin in the pathogenesis of orofacial pain. I believe that this kind of presentation of "key takeaways" is a very accessible form for the reader so I decided to remove repeated sentences from the text and move citations to Figure 3.
- "Figure 4 gives misleading information." - green down arrows correctly indicate reduced risk. Regarding the role of melatonin in sex hormones - contrary to many controversial studies in the past, the effect is significant on estrogen levels in women and there is no effect on testosterone in men. So it seems that the diagram is correct.
- I added a source to all figures.
- The title of section 7 has been changed to "A proposed cascade of events leading to orofacial pain and gender predisposition."
- The hypothetical patient is not my patient so I called him "hypothetical"
- I corrected some of the remarks regarding the English language myself, most of them has been corrected with MDPI native speaker editor..

All "marked yellow" sentences has also been corrected.

I changed last figure since Ive noticed mistake in "inflammation" word

Finally, I decided to add "....and depression" to the title and aditor changed it even further to: 

"Gut bless your pain—Roles of the gut microbiota, sleep, and melatonin in chronic orofacial pain and depression"

Once again, thank you very much for the reviews and I hope that this time the article will be much more pleasant to read.

Reviewer 2 Report

The authors state the role of gut microbiota,  sleep, and melatonin in the regulation of chronic orofacial pain. An unbalanced gut microbiome causes depletion of serotonin and melatonin and impacts the pathophysiology and treatment of chronic orofacial pain. The authors present their theory explaining the pathophysiology and support the hypothesis of the existence of a Gut-Sleep-Psycho-TMD axis.

It is a really interesting article based on extended bibliography which merits publication. There are also very nice schematic representations of this correlation . However, the article needs to be thoroughly revised for the English language by a native English speaker.

My opinion is to ACCEPt the article for publication after language editing.

Author Response

Dear reviewer,
Thank you for such in-depth analysis and constructive comments.
I will try to answer each comment.
- article selection criteria have been improved.
- I added a source to all figures.
- The title of section 7 has been changed to "A proposed cascade of events leading to orofacial pain and gender predisposition."
- I corrected some of the remarks regarding the English language myself, most of them has been corrected with MDPI native speaker editor..

All "marked yellow" sentences has also been corrected.

I changed last figure since Ive noticed mistake in "inflammation" word

Finally, I decided to add "....and depression" to the title and aditor changed it even further to: 

"Gut bless your pain—Roles of the gut microbiota, sleep, and melatonin in chronic orofacial pain and depression"

Once again, thank you very much for the reviews and I hope that this time the article will be much more pleasant to read.

Round 2

Reviewer 1 Report

The authors have done a good job responding to my comments on the previous version of their manuscript. I recommend the manuscript be accepted for publication after some minor English corrections regarding spelling mistakes throughout the text. Moreover, I would suggest that the authors consider using Figure 5 as a graphical abstract for this paper.

Author Response

Thank you for comments,

I have changed highlighted sentences and corrected minor spelling mistakes.

I believe there is no more to do since two independent native speakers from MDP revised and certified the paper.

Kind regards,

Lukas
